# Overexpression generates aberrant distribution of endocytic regulators - the case of the Rab11/LAMP1 compartment

Sarah Hornfeck, Evgeniya Trofimenko, Christian Widmann*

Department of Biomedical Sciences, University of Lausanne, Lausanne, Switzerland

☯ ET and C.W. are Joint Senior Authors
* christian.widmann@unil.ch

## Abstract

While the uptake of cargos via endocytosis and the subsequent trafficking through the cell is crucial for normal cellular function and tightly regulated, the study of this bears challenges. Most studies of Rab GTPases, the primary coordinators of endocytic progression, rely on ectopic expression of fluorescently tagged proteins via transient transfection. Previous studies already showed that the design of the fluorescent tag as well as the unpredictable nature of transient transfection can cause problems. Even though the pitfalls of overexpression have been reported for several research fields, the consequences of overexpression on endocytic trafficking are under-reported. To highlight the importance of working with endogenous levels of proteins to draw conclusions about endosome colocalization and identity, we present an example where the colocalization of two endosomal regulators/markers, Rab11 and LAMP1, varied drastically when these proteins were analyzed at their endogenous levels or following ectopic expression. When both proteins were ectopically expressed, up to 90% colocalization was observed. However, when analyzed at the endogenous level no colocalization was detectable. This study shows how important vesicular trafficking perturbation can occur following ectopic expression of endosomal proteins.

## Introduction

Vesicular trafficking is essential for proper cellular function. Transport through endocytic pathways, expression of cell surface components, secretion, recycling, and trafficking between organelles are coordinated by a family of small GTPases called Rab proteins, of which more than sixty members have been described so far. By cycling between GTP- and GDP-bound states [1], Rab proteins bind endosomal membranes and coordinate their transport across the cell through interactions with effector proteins, such as tethering proteins or motor proteins like kinesin and myosin [2,3]. The cycling of Rabs between GDP- (inactive) and GTP-bound (active) states is regulated

**Data availability statement:** All relevant data are within the manuscript and its Supporting Information files.

**Funding:** This work is supported by the Swiss National Science Foundation (grant n° 310030_207464).

**Competing interests:** The authors have declared that no competing interests exist.

by guanine nucleotide exchange factors (GEFs), which activate Rabs by facilitating the release of GDP, and GTPase-activating proteins (GAPs), which increase the intrinsic GTPase activity of Rab proteins, thus deactivating Rabs by promoting the hydrolysis of GTP to GDP. Upon GTP hydrolysis, the C-terminus—which is responsible for membrane interaction—is masked by GDP-dissociation inhibitors (GDIs), preventing Rabs from binding to membranes until the next activation event [4]. The trafficking of Rabs and their interaction with effector proteins enables the targeting of cargos such as transferrin, but also receptors and other cargos for recycling via Rab11 [5–7] and dextran for degradation in LAMP1-positive endosomes [8–10].

The development of fluorescently tagged proteins and their expression in cells has enabled researchers to investigate live cargo trafficking without the need for fixation and the laborious search for reliable antibodies with sufficient specificity and sensitivity. While ectopic expression of fluorescently tagged proteins permits live-cell cargo tracking and time-lapse experiments, the resulting overexpression may perturb the delicate balance of proteins regulating vesicular trafficking. This raises the question of the reliability of data obtained from overexpressed fluorescently tagged proteins. Ectopic expression can affect sensitive signaling processes that depend on precise protein concentrations [11]. Another potential problem results from the actual tagging of the protein of interest that can affect its functionality [12]. While the pitfalls of overexpression have been reported in areas such as plant biology [13], autophagy [14], and receptor interactions [15], they have not been extensively investigated in the context of vesicles and endosomal marker co-localization.

To highlight the importance of working with endogenous levels of proteins to draw conclusions about endosome colocalization and identity, we present an example where the colocalization of two endosomal regulators/markers, Rab11 and LAMP1, varied drastically when these proteins were analyzed at their endogenous levels or following ectopic expression. Rab11 controls vesicular recycling and LAMP1 is a lysosome and late endosome marker. When ectopically expressed, an artefactual compartment positive for these two proteins was observed. This compartment was extensive as up to 90% colocalization between Rab11 and LAMP1 could be detected. However, when Rab11 and LAMP1 were assessed at their endogenous levels, either by immunofluorescence or through generation of knock-in cell lines, no colocalization between them occurred. This study shows how important vesicular trafficking perturbation can occur following ectopic expression of endosomal proteins.

## Results

### Detection of a Rab11/LAMP1 vesicular compartment

To establish the relationship between recycling endosomes and late endosomes/lysosomes, two fluorescent protein-tagged markers for these vesicles (Rab11 and LAMP1, tagged with DsRed and GFP respectively) were overexpressed by transient transfection in HeLa cells and analyzed by confocal microscopy (Fig 1A). Intriguingly, a substantial colocalization between Rab11 and LAMP1 was observed when both proteins were overexpressed, with 52% of LAMP1 vesicles being positive for Rab11 and 75% of Rab11 vesicles also positive for LAMP1 (quantified in Fig 1E).

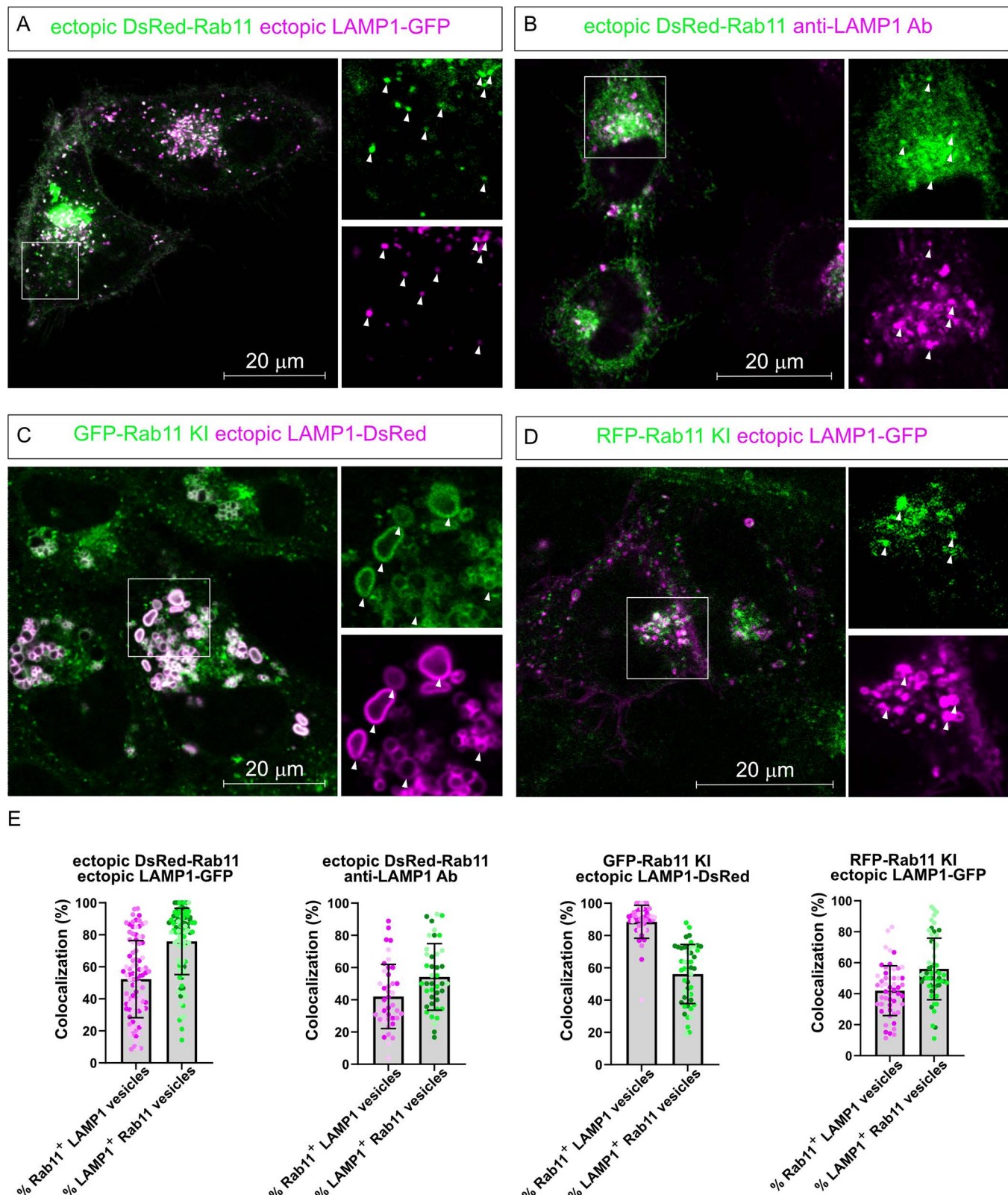

**Fig 1. Colocalization between LAMP1 and Rab11 based on different means of ectopic expression. (A)** Representative confocal image of HeLa wild-type cells transfected with both DsRed-Rab11 and LAMP1-GFP. **(B)** Representative confocal image of wild-type HeLa cells transiently transfected

with DsRed-Rab11 with subsequent immunofluorescence staining against LAMP1. **(C)** Representative confocal images of HeLa cells expressing endogenously tagged GFP-Rab11 transiently transfected with LAMP1-DsRed tagged vector. **(D)** Representative confocal images of RFP-Rab11 KI cells transfected with LAMP1-GFP. The scalebars indicate 20 µm and the zoomed areas are indicated by squares. Arrowheads indicate the structures where colocalization occurs. **(E)** Quantitation of the colocalization between Rab11 and LAMP1 in the different experimental conditions shown in panels A-D (at least 40 cells per condition were scored from three independent experiments indicated by different color shades; 1403 Rab11 vesicles and 2265 LAMP1 vesicles were counted for condition A, 904 Rab11 vesicles and 908 LAMP1 vesicles were counted for condition B; 1541 Rab11 vesicles, and 965 LAMP1 vesicles were counted for condition C; 1427 Rab11 vesicles and 1653 LAMP1 vesicles were counted for condition **D)**. The bars in the graph represent the mean and the error bars correspond to the standard deviation.

An endosomal compartment positive for both markers could have represented a novel intermediate between recycling and degradation. As a first test to verify this colocalization, immunofluorescence staining for LAMP1 using a validated antibody (Figs 2A,B) was performed in cells overexpressing dsRed-Rab11 (Fig 1B). This showed again a marked colocalization between Rab11 and LAMP1 (42% of LAMP1 vesicles positive for Rab11 and 54% of Rab11 vesicles positive for LAMP1). To further investigate whether this observation was replicable when one of these proteins was assessed at its endogenous level, we generated stable cell lines in which GFP or RFP was knocked in the endogenous locus of Rab11 (Fig 2A,B). Ectopic expression of LAMP1-DsRed1 in the GFP-Rab11 knock-in (KI) cells led to large vesicles positive for both proteins (88% of LAMP1 vesicles positive for Rab11 and 56% of Rab11 vesicles positive for LAMP1) (Fig 1C). Such colocalization was already observed at the lowest DNA amount used for transfection (0.25 µg) at which fluorescent signal was observed for LAMP1. In this case 77% of LAMP1 vesicles were positive for Rab11 and 53% of Rab11 vesicles were positive for LAMP1 (18 cells analyzed) (S3 Fig). Similarly, 42% of LAMP1 vesicles were positive for Rab11 and 56% of Rab11 vesicles were positive for LAMP1 after ectopic expression of LAMP1-GFP in RFP-Rab11 KI cells (Fig 1D). The substantial variations in the extent of colocalization between Rab11 and LAMP1 in the different experiment settings used in Fig 1 indicate that the use of different fluorescent proteins and mode of protein expression impact the formation of Rab11/LAMP1 vesicles but otherwise were suggestive of the existence of a newly identified compartment at the interface of recycling Rab11 vesicles and LAMP1 endocytic vesicles.

### The Rab11/LAMP1 compartment cannot be detected at endogenous Rab11 and LAMP1 levels

In Fig 1A,C,D, at least LAMP1 was ectopically expressed. To assess whether the Rab11/LAMP1 compartment could be detected when both proteins were assessed at endogenous levels, two approaches were chosen. First, immunofluorescence staining for LAMP1 was performed in the GFP-Rab11 KI cells with a validated anti-LAMP1 antibody (Fig 3A). In this setting, Rab11 and LAMP1 showed distinct cellular patterns and did not colocalize at all. The second approach was to knock-in mKusabiraOrange2 in the LAMP1 locus (Fig 2E,F) in GFP-Rab11 KI cells, therefore generating a double-KI cell line in which both Rab11 and LAMP1 were tagged with a distinct fluorescent protein (Fig 3B). Confocal imaging of the double KI cell line revealed no colocalization between Rab11 and LAMP1 (quantified in Fig 3C), further corroborating that Rab11 and LAMP1 are not found on the same vesicles in physiological conditions.

### Discussion

While the initially observed high colocalization between LAMP1 and Rab11 suggested the existence of an intriguing new vesicular compartment, subsequent experiments in cells expressing these proteins at the endogenous levels revealed the artefactual existence of this Rab11/LAMP1 compartment. This raises the general concern that overexpression may be inappropriate for the study of vesicular trafficking. This notion is further corroborated as this artefactual colocalization was also observed in epithelial and non-cancerous cell lines (S4 Fig). Although several papers have already warned about the unpredictable nature of transient transfections [11] and the importance of careful consideration of fluorophore tag positioning and choice [12,16,17], problems associated with ectopic expression of vesicular markers have been under-reported. Our work, by highlighting an example where ectopic expression of vesicular proteins create artefactual compartments,

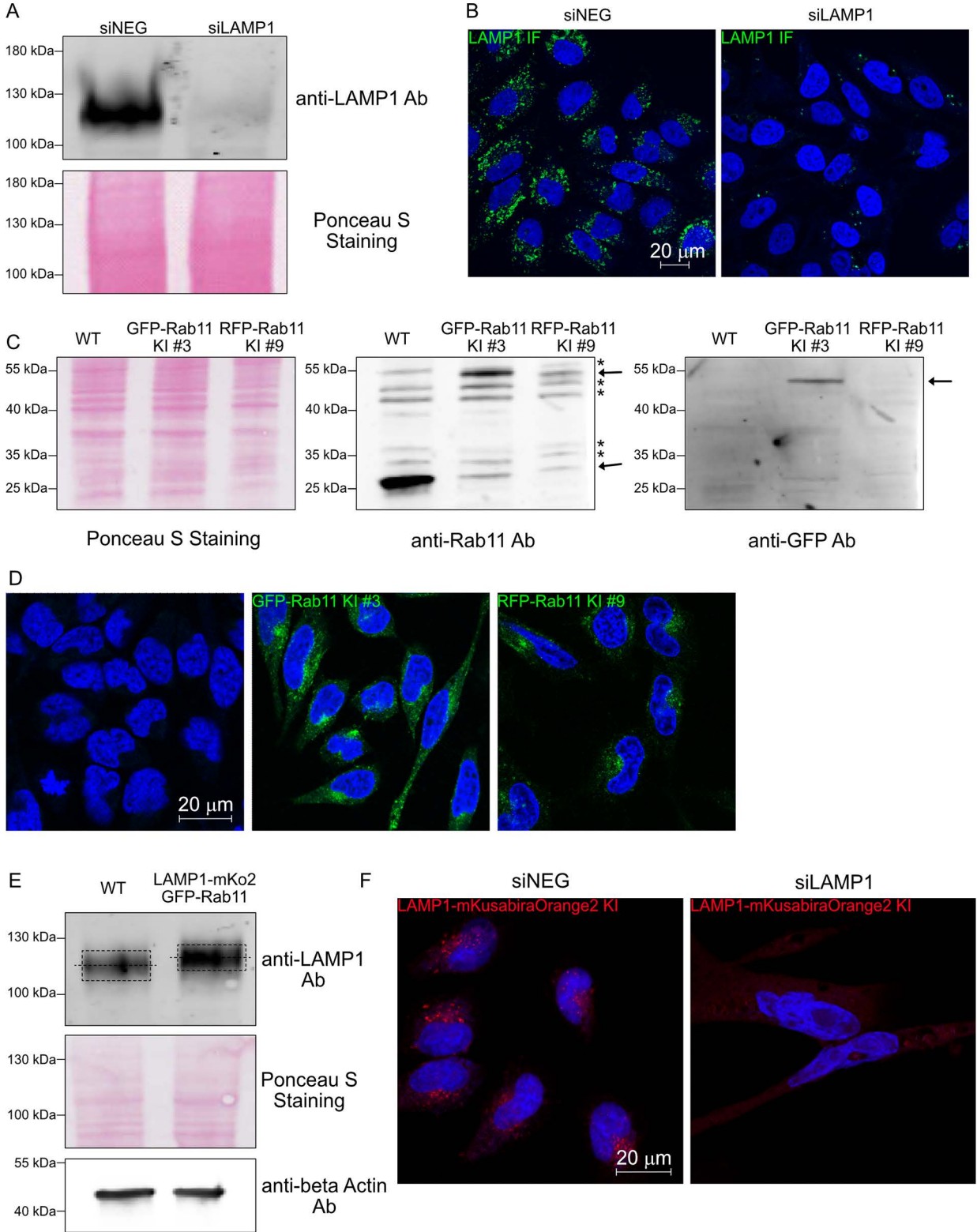

**Fig 2. Validation of the anti-LAMP1 antibody and the knock-in cells (A)** Western-blot validation of the LAMP1 antibody by comparing cells treated with a control siRNA and an siRNA directed at LAMP1. **(B)** Representative confocal images of LAMP1-stained HeLa wild-type cells treated with a control siRNA or an siRNA directed at LAMP1. **(C)** Western-blot comparison of wild-type HeLa cells vs. GFP-Rab11 knock-in cells (clone #3) and

RFP-Rab11 knock-in cells (clone #9) using an anti-Rab11 antibody (middle panel) as well as an anti-GFP antibody that recognises GFP but not RFP to visualize GFP-KI Rab11 (right panel). Ponceau S Staining is shown to evaluate the evenness of loading (left panel). The asterisks show unspecific bands and the arrows point to Rab11. **(D)** Representative confocal images of wild-type HeLa cells, the GFP-Rab11 knock-in cells (clone #3) and the RFP-Rab11 knock-in cells (clone #9). **(E)** Western-blot comparison of wild-type HeLa cells vs. LAMP1-mKusabiraOrange2 and GFP-Rab11 double knock-in cells using an anti-LAMP1 antibody. Whole protein loading is shown (Ponceau S Staining) as well as an anti-β-actin staining. **(F)** Representative confocal images of LAMP1-stained LAMP1-mKusabiraOrange2 knock-in cells treated with a control siRNA or an siRNA directed at LAMP1. Scalebars: 20 μm.

indicates that investigation of vesicular trafficking in cells can be seriously perturbed when the balance of the regulators of this trafficking is altered. We therefore advocate that key experiments are validated in conditions where the proteins investigated in the context of vesicular trafficking are studied under endogenous level conditions.

## Materials and methods

### Cell culture

HeLa cells (ATCC, CCL-2, RRID:CVCL_0030) were cultured in RPMI1640 medium (Gibco, 61870−010) supplemented with 10% fetal bovine serum (Gibco, A5256701) in tissue culture (TC)-treated 10 cm dishes (Corning, 430167) at 37°C and 5% $CO_2$ (Salvislab Biocenter). HEK293T (ATCC, CRL-3216, RRID:CVCL_0063), and U2OS (ATCC, HTB-96, RRID:CVCL_0042) cells were cultured in DMEM (Gibco, 61965026) supplemented with 10% fetal bovine serum (Gibco, A5256701) in TC-treated 10 cm dishes (Corning, 430167) at 37°C and 5% $CO_2$ (Salvislab Biocenter). The cells were passaged at ~80% confluency using trypsin-EDTA (Gibco, 2534416) to a new 10 cm dish.

### Antibodies

The anti-LAMP1 antibody (BD Bioscience, 555798, RRID:AB_396132) was used at 1:200 for immunofluorescence and 1:1000 for western blot (WB). The anti-Rab11 antibody (Invitrogen, 71–5300, RRID:AB_2533987) was used at 1:1000 for WB. The anti-GFP antibody (Roche, 11814460001, RRID:AB_390913) was used at 1:1000 for WB. The anti-β-Actin antibody (Cell Signaling Technology, 4970, RRID:AB_2223172) was used at 1:1000 for WB. The FITC conjugated anti-mouse antibody (Jackson Immunoresearch, 115-095-146, RRID:AB_2338599) was used at 1:500 for immunofluorescence. The AlexaFluor647 conjugated anti-mouse antibody (Invitrogen, A31571, RRID:AB_162542) was used at 1:500 for immunofluorescence. The anti-mouse, AlexaFluor680 conjugated secondary antibody (Invitrogen, A21057, RRID:AB_2535723) with the anti-rabbit, AlexaFluor800 conjugated secondary antibody (Invitrogen, A32808, RRID:AB_2762837) or the anti-rabbit, AlexaFluor680 conjugated secondary antibody (Invitrogen, A21109, RRID:AB_2535758) with the anti-mouse, AlexaFluor800 conjugated secondary antibody (Invitrogen, A32730, RRID:AB_2633279) were used at a dilution of 1:5000 for WB.

### Plasmids

The plasmid expressing DsRed-hRab11 (N-terminal fluorescent tag) was purchased from Addgene (12679, RRID:Addgene_12679), as well as the plasmid encoding hLAMP1-mGFP (34831, RRID:Addgene_34831) (C-terminal fluorescent tag). The plasmid encoding hLAMP1-DsRed was produced by standard subcloning methods, using the PspOMI and NotI restriction enzymes, and inserting LAMP1 from the LAMP1-mGFP plasmid into a backbone encoding dsRed1 for C-terminal tagging (Addgene, 13386, RRID:Addgene_13386).

### Generation of stable knock-In (KI)

Stable fluorescent knock-ins were generated using the Crispr/Cas9 based ChoP-In approach [18]. Briefly, a plasmid containing Cas9 with sgRNAs was generated in a pX330 vector (Addgene, 42230, RRID:Addgene_42230). sgRNA was designed to result in a double stranded break at the first amino acid at the N-terminus of Rab11 (Table 1). The donor PCR

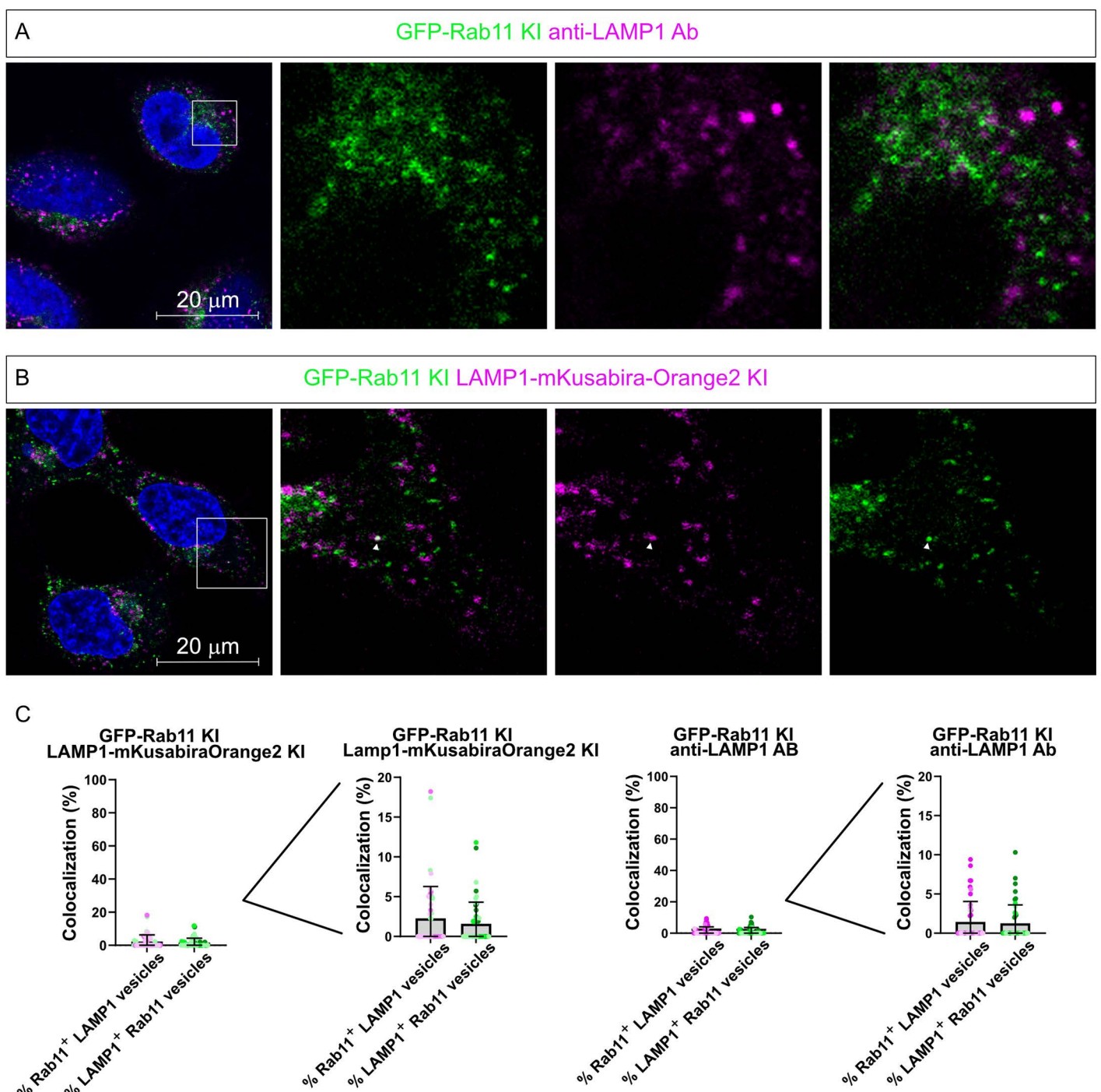

**Fig 3. Absence of colocalization between LAMP1 and Rab11 when studied at their endogenous levels. (A)** Representative confocal image of immunofluorescence staining using an anti-LAMP1 antibody (visualized with an AlexaFluor647-conjugated secondary antibody) in GFP-Rab11 cells. **(B)** Representative confocal image of LAMP1-mKusabiraOrange2 / GFP-Rab11 knock-in cells. Scalebars: 20 μm. Enlarged areas are indicated by squares. Arrowheads indicate the structures where colocalization occurs. **(C)** Quantitation of the colocalization between Rab11 and LAMP1 in the different experimental conditions shown in panels A-B (at least 45 cells per condition were scored from three independent experiments; 1809 Rab11 vesicles and 1227 LAMP1 vesicles were counted for condition A; 1266 Rab11 vesicles and 1245 LAMP1 vesicles were counted for condition B). Scale enlargement is also shown to better appreciate the very low colocalization between Rab11 and LAMP1. The bars in the graph represent the mean and the error bars correspond to the standard deviation..

**Table 1. sgRNAs used for the generation of stable Kis.**

| | |
|---|---|
| sgRNA targeting Rab11 FWD | CAC CGT TCG CTC CTC GGC CGC GCA A |
| sgRNA targeting Rab11 REV | AAA CTT GCG CGG CCG AGG AGC GAA C |
| sgRNA targeting LAMP1 FWD | CAC CGG TGC ACC AGG CTA GAT AGT C |
| sgRNA targeting LAMP1 REV | AAA CGA CTA TCT AGC CTG GTG CAC C |

product contained the selected fluorophores (EGFP, TagRFP or mKusabiraOrange2) flanked by the sgRNA sequences. Lamp1 was endogenously tagged at the C-terminus. To achieve this, an anti-sense sgRNA was designed resulting in deletion of the last two amino acids to avoid stop codon generation.

Both the PCR donor and Cas9 plasmid with sgRNA were transiently transfected in 200'000 HeLa wild-type cells following the manufacturer's instructions using Lipofectamine2000 (Invitrogen, 11668019) at a mass ratio of 1:1 in 6-well plates. 48h after transfection the cells were sorted (BD FACSAria II SORP) into 96-well plates (Corning, 3599) to obtain single-cell clones. Once the clones expanded, they were checked for the integration of the fluorescent tag by Western blot and confocal imaging.

## Western blotting

Cells were washed with ice-cold PBS and lysed in the Cell signaling lysis buffer (Cell Signalling Technology, 9803) incubated for 5 min on ice. The cells were scrapped, vortexed and centrifuged at 14'000xg at 4 °C for 10 min (Eppendorf, 5415 R). The supernatant was collected and the protein concentration measured using a BCA Protein Assay Kit (ThermoFischer, A55864). 20 µg protein were loaded onto pre-casted SDS gels (Bio-Rad, 4561093) and run at 200 V for 30–45 min. The proteins were transferred to a nitrocellulose membrane, followed by ponceau staining used as a visual loading control. The membrane was blocked using 3% BSA in TBS-tween and the primary antibodies incubated overnight at 4 °C diluted the same buffer while rocking. The membrane was washed three times with TBS-tween and incubated with the secondary antibody for 1h at room temperature with gentle rocking. The membrane was imaged using the Li-Cor Odyssey 9120 and analyzed using FIJI ImageJ 1.54p [19].

## Immunofluorescence staining

Cells were seeded in 6-well plates (Corning, 3516) with sterile coverslips (VWR, ECN631–1577) to 80% confluency. Cells were fixed with 4% PFA for 20 min. After washing, the cells were incubated in blocking buffer containing 3% BSA, 0.025% saponin, and 1.5% glycine in PBS for 45 min at room temperature. The primary antibody was incubated in an antibody incubation buffer (0.1% BSA and 0.01% saponin in PBS) for two hours at room temperature. After washing with PBS three times, the secondary antibody was diluted in the same antibody incubation buffer and incubated for one hour at room temperature protected from light. Subsequently the coverslips were washed with PBS thrice and rinsed with water before being mounted with Vectashield Vibrance mounting medium with DAPI (Vector Laboratories, H-1800).

## Live imaging

HeLa cells were seeded at a concentration of 200'000 cells per well in 35 mm glass bottom plates (MatTek Corporation, P35G-1.5-14-C). Depending on the experiment the cells were imaged with or without Hoechst 33342 (Invitrogen, H3570). The cells were imaged using a Leica Stellaris 8 microscope with an HC PL APO 63x/1.40 oil CS2 objective, equipped with 405 nm DMOD, 488 nm, 561 nm and 638 nm lasers and HyD S detectors. For Hoechst or DAPI visualization, excitation was performed with a 405 nm laser and emission was recorded using a 415 nm – 486 nm filter. For GFP visualization, excitation was performed with a 488 nm laser and emission was recorded using a 493 nm – 561 nm filter. For DsRed, RFP, and mKusabiraOrange2, excitation was performed with a 561 nm laser and emission was recorded using a 566 nm – 637 nm

filter. The images were acquired at 0.09 µm pixel size with two times line averaging and a Leica K5 sCMOS 80%QE, 2048×2048, 6.5×6.5 µm pixels camera. Acquisition parameters such as gain and laser intensity were kept consistent between the replicates. The pixel dwell time was <0.5 µs. Controls ensured there was no bleed-through between the channels (Panel A and B S2 Fig) and the imaging plane was chosen as shown in Panel C in S2 Fig.

## Transient transfection

Two hundred thousand cells were seeded onto 35 mm glass bottom dishes or on glass coverslips in a 6-well plate and were allowed to adhere overnight. The transient transfections were performed using Lipofectamine2000 (Invitrogen, 11668019). For DsRed-hRab11, 1 µg was used, for hLAMP1-mGFP 0.5 µg and for hLAMP1-DsRed 0.5 µg. The DNA amount used was titrated as shown for the example of LAMP1-DsRed in GFP-Rab11 KI cells in S3 Fig. Media was changed 24 hours post transfection. Cells were further processed for either live visualization or immunostaining.

## Colocalization analysis

The colocalization analysis was visually assessed using the LAS X Office software (Leica Microsystems, version 1.4.5.27713). The total number of vesicles was counted for all markers which were to be compared. To determine the percentage of colocalization, the number of vesicles positive for both markers was determined. Two marker colocalize when the signals overlapped and matched in shape precisely in the confocal images. This is also described in detail elsewhere [20]. Lastly, the percentage of colocalization was determined by dividing the colocalizing vesicles by the total number of vesicles for the given marker. The manual quantitation was validated by Manders and Pearsons coefficients using the JaCop plugin [21] in FIJI performed in the same set of images as the manual quantitation (Panel D in S2 Fig). The threshold for Manders colocalization efficient was set to visualize endosomes without cytosolic signal for each cell. All graphs were generated using GraphPad Prism 10.5.0 for Windows (GraphPad Software).

## siRNA treatment

The siPOOLs were ordered from siTOOLS Biotech and diluted to 150 nM in water and stored at −20°C. HeLa cells were seeded with a concentration of 75'000 cells on either 35 mm glass bottom plates (MatTek Corporation, P35G-1.5-14-C) for live imaging or in 6-well plates (Corning, 3516) with sterile coverslips (VWR, ECN631–1577) for IF and without coverslips for WB. Before the cells attached, the siRNA was transfected using Lipofectamine RNAiMAX (Invitrogen, 13778075) following the vendors instructions with a final concentration of 3 nM of siRNA. 24h later, the KD was repeated by transfecting the cells again with the siRNAs using Lipofectamine2000 (Invitrogen, 11668019) and the siRNA incubated for another 48h for a total of 72h. Subsequently, the cells were either imaged with the confocal microscope or processed for IF or WB analysis.

## Supporting information

**S1 Fig. Raw images of the Western blots.**
(PDF)

**S2 Fig. Additional controls.**
(PDF)

**S3 Fig. Titration of DNA for ectopic expression.**
(PDF)

**S4 Fig. Colocalization of ectopic Rab11 and LAMP1 in additional cell lines.**
(PDF)

 

**S5 Table. Raw data.**
(XLSX)

## Acknowledgments

We are thankful to the Cellular Imaging Facility and the Flow Cytometry Facility at the University of Lausanne for the resources provided and their technical help.

## Author contributions

**Conceptualization:** Christian Widmann, Sarah Hornfeck.

**Data curation:** Christian Widmann, Sarah Hornfeck.

**Formal analysis:** Christian Widmann, Sarah Hornfeck.

**Funding acquisition:** Christian Widmann.

**Investigation:** Sarah Hornfeck.

**Project administration:** Christian Widmann.

**Supervision:** Christian Widmann, Evgeniya Trofimenko.

**Validation:** Christian Widmann.

**Writing – original draft:** Sarah Hornfeck.

**Writing – review & editing:** Christian Widmann, Sarah Hornfeck, Evgeniya Trofimenko.

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
