## [Decision Letter · Decision Letter 0]

9 Dec 2025

PONE-D-25-49367Overexpression generates aberrant distribution of endocytic regulators - the case of the Rab11/LAMP1 compartmentPLOS One

Dear Dr. Widmann,

Thank you for submitting your manuscript to PLOS ONE. After careful consideration, we feel that it has merit but does not fully meet PLOS ONE’s publication criteria as it currently stands. Therefore, we invite you to submit a revised version of the manuscript that addresses the points raised during the review process.

We look forward to receiving your revised manuscript.

Kind regards,

Prasanth Puthanveetil

Academic Editor

PLOS One

Journal Requirements:

[This work is supported by the Swiss National Science Foundation (grant n° 310030_207464).].

[The laboratory of CW is supported by the Swiss National Science Foundation (grant n° 310030_207464). We are thankful to the Cellular Imaging Facility at the University of Lausanne for the resources provided and their technical help.]

[This work is supported by the Swiss National Science Foundation (grant n° 310030_207464).]

Additional Editor Comments:

Dear Authors,

Please address the concerns raised by our reviewers and submit your revised version within the allocated time.

Looking forward to your submission.

Thanks,

Handling Editor.

Reviewers' comments:

Reviewer's Responses to Questions

**Comments to the Author**

1. Is the manuscript technically sound, and do the data support the conclusions?

Reviewer #1: Yes

Reviewer #2: Yes

Reviewer #3: Yes

Reviewer #4: Yes

Reviewer #5: Partly

2. Has the statistical analysis been performed appropriately and rigorously? 

Reviewer #1: Yes

Reviewer #2: N/A

Reviewer #3: Yes

Reviewer #4: N/A

Reviewer #5: No

3. Have the authors made all data underlying the findings in their manuscript fully available?

Reviewer #1: Yes

Reviewer #2: Yes

Reviewer #3: Yes

Reviewer #4: Yes

Reviewer #5: Yes

4. Is the manuscript presented in an intelligible fashion and written in standard English?

Reviewer #1: Yes

Reviewer #2: Yes

Reviewer #3: Yes

Reviewer #4: Yes

Reviewer #5: Yes

5. Review Comments to the Author

Reviewer #1: In the current report, Hornfeck et al. studied the subcellular distribution of Rab11 and Lamp1, known to reside in the recycling endosome and late endosome/lysosome, respectively. Not quite surprisingly, the authors found that ectopic overexpression of these proteins, tagged with fluorescent markers, affects their distribution, leading to wrong conclusions about their subcellular colocalization. When both proteins were knocked in to allow endogenous expression, no colocalization was observed.

While the data provided seems rather convincing, the overall information is too limited to justify publication as a Research Article or even Qualitative Research. The authors are encouraged to utilize the double knock-in cells to determine questions regarding these interesting organelles.

Reviewer #2: A clear, concise and thorough article that used logical scientific approaches to highlight an important methodological concern in trafficking research. Although it is widely accepted that ectopic protein expression leads to abnormally high protein expression levels and likely alters functional pathways, evidence for this is very rarely published likely due to negative results bias. This article provides novel citable evidence in a commonly used cell model in the field. Publication of this data can benefit relevant groups when designing experiments and considering results as well as prevent other research groups from falsely identifying this now discredited intermediate vesicular compartment.

An appropriate number of experimental combinations for ectopic and endogenous expression levels was performed to support their conclusion. Strong evidence with robust correlation between images and quantification. Very clear explanation of results and methods which could be easily reproduced. No statistical analysis, but can't identify any need within this study.

Although the evidence here shows ectopic protein expression can have negative consequences, it is common practice to titrate the amount of DNA in ectopic expression experiments to try and achieve less abnormal expression patterns. Optimisation of the DNA concentrations to reduce aberrant colocalization would increase the impact of this research, 2ug is high. Minimally, address this in the article text as a limitation.

Suggested improvements:

- Fig 1E and 3C are not referred to in text about cololocalisation quantification, adding this would be beneficial.

- Line 95: 'with when' remove with

- Changing the figure order to have 2 C/D above 2A/B would be beneficial to the reader in order of how it is read.

Recommended for acceptance after these minor revisions.

Reviewer #3: The authors address the relevance of studying vesicular trafficking at endogenous level and how the overexpression of Rab11 and LAMP1 generates a colocalization that was not present in an endogenous context. Besides, they showed how the approach of an overexpression by using tagged proteins could produce artefactual compartments. Although is a very interesting work, there are some controls and questions that should be addressed.

1.- Lane 52, include more recent works in Rab11 trafficking and recycling.

2.- Lane 53, include recent works for LAMP1 context.

3.- Lane 84, include the name of the two fluorescent proteins in this part of the text.

4.- Lane 86, all Figure 1, include the nuclear staining, either DAPI or Hoechst.

5.- Lane 91, Figure 2C, obtain a housekeeping signal for the Western Blot results, and a contour for each wb result will be fine.

6.- Lane 96, Figure 2A, include a housekeeping signal for the Western Blot result, I suggest repeating the whole Western Blot by running the protein extracts in the same gel, thus, getting compared signal results.

7.- Lane 332, Figure 3C, look for a better representation of the points in the graphic, because is barely appreciate, most of them are enclose in the 20% range, so, a better distribution of them should be addressed for a better analysis.

8.- Lane 331, figure 2A, get the signal by using a GFP antibody in the Western Blot for the fusion proteins.

9.- Do the authors think that by using an inducible system under the control of Doxocycline for example, it will be possible to reach good results by using tagged proteins? For vesicular traffic studies?

Reviewer #4: S. Hornfeck et al. present an interesting and important manuscript, as it highlights the often overlooked risk of creating artifacts through the overexpression of proteins and/or through tagging them with fluorophores like GFP.

They set out to investigate the role of Rab11 in the internalization, recycling or degradation processes. They convincingly demonstrate that, when investigating this question using the common approach of overexpressing labelled fusion proteins, artifacts can arise that do not reflect the situation of these proteins under physiological expression levels. Upon overexpression they observed colocalization of Rab11 and LAMP1 in many vesicles, suggesting the existence of an intermediate compartment of recycling endosomes and lysosomes. However, they did not observe this at physiological expression levels.

In general, the experiments were well conducted and included appropriate controls. However, a few points should be addressed:

P12, line 110: clarify in which of the images in Fig. 1 LAMP-1 is ectopically expressed, as it is not the case in Fig.1B

Page 12, line 115: should be GFP-Rab11 KI (makes more sense and is so written in the Fig. 3B)

Page 12, line 117 Transfer “Figure 2E” to line 115 after “Lamp1 locus” as Figure 2E does not show a double expressing cell line.

Page 12, line 117. Replace “Figure 2E” by “Figure 3B”. “Figure 3B” can now be removed from line 118.

Page 16, lines 192 -194: this is not a full sentence, verb is missing (contained?)

Page 16, lines 201: word is missing

Among the antibodies listed in “Materials and Methods”, it is not clear for all of them where they were used:

In Fig. 1B the FITC-anti-mouse obviously was used in combination with ectopic DsRed-Rab11 for labelling of endogenous LAMP1. Where was the listed AlexaFluor647 conjugated anti-mouse antibody used at 1:500 for immunofluorescence (page 16, line 175)?

Page 16, line 173: “The Cy5 conjugated anti-Rabbit antibody was used at 1:500 for immunofluorescence“. Where? It would indeed have been even more convincing if both endogenous proteins Rab11 and LAMP-1 - without any fused fluorescent tags - had been detected only by antibodies. That might have been the right antibody to do it.

Fig. 1: Could the authors comment on why the vesicles in C and D look so different, even though the main difference is only in the colour of the fluorophores. Off-target effects? Did all clones look like this?

Page 17, line 201: Could the authors elaborate on how many cells successfully replaced the wild type with the fusion protein and whether the phenotype was affected through off-target effects. Which clones where selected for the experiments? If western blot was standard procedure, why isn’t there one shown for the LAMP1-mKusabiraOrange2 KI in Fig. 2?

Reviewer #5: General Comments

This manuscript addresses an important methodological issue in cell biology—the widespread reliance on transient overexpression of fluorescently tagged endosomal markers. By systematically comparing the colocalization patterns of Rab11 and LAMP1 under overexpression versus endogenous expression, the authors demonstrate that high levels of ectopic expression can generate artefactual endosomal compartments.

However, the study relies almost entirely on qualitative imaging and lacks sufficient quantitative and mechanistic depth to fully substantiate its conclusions. Below, I outline major and minor points that should be addressed before publication.

Major Comments

1. Quantitative Colocalization Analysis

• The quantification of Rab11/LAMP1 overlap is currently based on manual visual scoring. This approach is inherently subjective and insufficient for publication.

• The authors should apply standard quantitative methods such as Pearson’s correlation coefficient, Manders’ coefficients, or object-based colocalization analysis, with proper controls.

• Include GFP/DsRed-only controls to exclude bleed-through and aggregation artefacts.

• Provide the number of cells and vesicles analyzed, the statistical tests applied, and the variation across replicates (e.g., standard deviation or SEM).

2. Mechanistic Explanation for Artefactual Colocalization

• The manuscript convincingly shows that overexpression causes Rab11 and LAMP1 to colocalize but does not explore the underlying mechanism.

• The authors should at least discuss possible explanations, such as:

o Overexpression saturating Rab effectors or competing for membranes.

o Altered membrane identity due to overloading of Rab GTPases.

o Non-physiological aggregation of tagged proteins.

• Would using alternative or weaker promoters that reduce expression levels mitigate this artefact?

• Does the Rab11/LAMP1-positive compartment have a functional impact on cargo trafficking?

3. Discussion Depth and Context

The discussion reiterates known concerns about overexpression but could go further. The authors should relate their findings to Rab cascade models (Rab5–Rab7–Rab11 transitions) and mechanisms of endosomal identity maintenance. It would also be valuable to highlight how this artefact may have influenced previous studies and to discuss alternative approaches to overcome these limitations.

4. Cell-Type Specificity

All experiments are performed in HeLa cells. A note on whether this artefact generalizes to other cell types (e.g., epithelial or neuronal cells) would strengthen the study’s impact.

Minor Comments

1. Replace phrases such as “a novel compartment” with “an artefactual compartment generated by overexpression” throughout the text.

2. Specify whether Rab11 and LAMP1 constructs were N- or C-terminally tagged.

3. In the Methods section, specify:

a. How vesicles were identified (thresholding criteria, size filters).

b. Image acquisition parameters (z-step size, laser power, pixel dwell time, gain).

4. Ensure consistent gene/protein notation (e.g., Rab11A vs. Rab11).

6. PLOS authors have the option to publish the peer review history of their article (what does this mean? ). If published, this will include your full peer review and any attached files.). If published, this will include your full peer review and any attached files.

**Do you want your identity to be public for this peer review?** For information about this choice, including consent withdrawal, please see our For information about this choice, including consent withdrawal, please see our Privacy Policy ..

Reviewer #1: No

Reviewer #2: No

Reviewer #3: No

Reviewer #4: No

Reviewer #5: **Yes:** Bhuvanasundar RanganathanBhuvanasundar Ranganathan

---

## [Author Response · Author response to Decision Letter 1]

23 Jan 2026

Please refer to our Rebuttal letter that we have uploaded.

---

## [Decision Letter · Decision Letter 1]

16 Mar 2026

Overexpression generates aberrant distribution of endocytic regulators - the case of the Rab11/LAMP1 compartment

PONE-D-25-49367R1

Dear Dr. Widmann,

We’re pleased to inform you that your manuscript has been judged scientifically suitable for publication and will be formally accepted for publication once it meets all outstanding technical requirements.

Kind regards,

Pirkko L. Härkönen, M.D., Ph.D.

Academic Editor

PLOS One

Additional Editor Comments (optional):

Reviewers' comments:

Reviewer's Responses to Questions

**Comments to the Author**

1. If the authors have adequately addressed your comments raised in a previous round of review and you feel that this manuscript is now acceptable for publication, you may indicate that here to bypass the “Comments to the Author” section, enter your conflict of interest statement in the “Confidential to Editor” section, and submit your "Accept" recommendation.

Reviewer #1: All comments have been addressed

Reviewer #2: All comments have been addressed

Reviewer #4: All comments have been addressed

Reviewer #5: All comments have been addressed

2. Is the manuscript technically sound, and do the data support the conclusions?

Reviewer #1: Yes

Reviewer #2: Yes

Reviewer #4: (No Response)

Reviewer #5: Yes

3. Has the statistical analysis been performed appropriately and rigorously? 

Reviewer #1: Yes

Reviewer #2: Yes

Reviewer #4: (No Response)

Reviewer #5: Yes

4. Have the authors made all data underlying the findings in their manuscript fully available?

Reviewer #1: Yes

Reviewer #2: Yes

Reviewer #4: (No Response)

Reviewer #5: Yes

5. Is the manuscript presented in an intelligible fashion and written in standard English?

Reviewer #1: Yes

Reviewer #2: Yes

Reviewer #4: (No Response)

Reviewer #5: Yes

6. Review Comments to the Author

Reviewer #1: The authors successfully addressed my concerns and in its resent form the manuscript meets Plos One merit.

Reviewer #2: The authors have addressed all my comments from the first review and I now reccomend that it should be published.

Reviewer #4: (No Response)

Reviewer #5: I appreciate the authors for carefully considering the reviewers’ comments and addressing them appropriately.

7. PLOS authors have the option to publish the peer review history of their article (what does this mean? ). If published, this will include your full peer review and any attached files.). If published, this will include your full peer review and any attached files.

**Do you want your identity to be public for this peer review?** For information about this choice, including consent withdrawal, please see our For information about this choice, including consent withdrawal, please see our Privacy Policy ..

Reviewer #1: No

Reviewer #2: No

Reviewer #4: No

Reviewer #5: **Yes:** Bhuvanasundar RenganathanBhuvanasundar Renganathan

---

## [Editor Report · Acceptance letter]

PONE-D-25-49367R1

PLOS One

Dear Dr. Widmann,

I'm pleased to inform you that your manuscript has been deemed suitable for publication in PLOS One. Congratulations! Your manuscript is now being handed over to our production team.

Kind regards,

on behalf of

Dr. Pirkko L. Härkönen

Academic Editor

PLOS One